# Growth Hormone-Releasing Hormone (GHRH) Antagonist Peptides Combined with PI3K Isoform Inhibitors Enhance Cell Death in Prostate Cancer

**DOI:** 10.3390/cancers17101643

**Published:** 2025-05-13

**Authors:** Carlos Perez-Stable, Alicia de las Pozas, Medhi Wangpaichitr, Wei Sha, Haibo Wang, Renzhi Cai, Andrew V. Schally

**Affiliations:** 1Research Service, Bruce W. Carter Veterans Affairs Medical Center, Miami, FL 33125, USA; delaspozasalicia@gmail.com (A.d.l.P.); mwangpaichitr@med.miami.edu (M.W.); wsha@med.miami.edu (W.S.); renzhi1938@gmail.com (R.C.); aschally@med.miami.edu (A.V.S.); 2Geriatric Research, Education, and Clinical Center, Bruce W. Carter Veterans Affairs Medical Center, Miami, FL 33125, USA; 3Division of Gerontology & Palliative Medicine, Department of Medicine, University of Miami Miller School of Medicine, Miami, FL 33136, USA; 4Sylvester Comprehensive Cancer Center, University of Miami Miller School of Medicine, Miami, FL 33136, USA; 5Department of Surgery, Division of Cardiothoracic Surgery, University of Miami Miller School of Medicine, Miami, FL 33136, USA; 6South Florida VA Foundation for Research and Education, Miami, FL 33125, USA; 7Interdisciplinary Stem Cell Institute, University of Miami Miller School of Medicine, Miami, FL 33136, USA; 8Department of Medicine, University of Miami Miller School of Medicine, Miami, FL 33136, USA; 9Department of Pathology, University of Miami Miller School of Medicine, Miami, FL 33136, USA

**Keywords:** prostate cancer, growth hormone-releasing hormone antagonist, PI3K, androgen receptor, AKT, ERK, Mcl-1

## Abstract

The increased use of potent androgen receptor antagonists has resulted in a rise in advanced prostate cancers resistant to androgen deprivation therapy with few treatment options. GHRH expression in cancers has led to the development of peptide antagonists (e.g., MIA-602 and -690) for therapeutic treatment. However, MIA-602/690 GHRH antagonists alone are not likely to be effective against advanced prostate cancer. We identified the novel combination of PI3K inhibitors + MIA-602/690 that increased cell death in all types of prostate cancer cells, including ones resistant to androgen deprivation therapy. The ability of MIA-602/690 and PI3K inhibitors to affect multiple signaling pathways may enhance cell death and optimize therapeutic benefit.

## 1. Introduction

Prostate cancer (PCa) is initially responsive to androgen deprivation therapy (ADT) but can develop resistance, leading to the progression of castration-resistant PCa (CRPC) [1,2]. New and more potent ADT drugs (e.g., androgen receptor [AR] antagonist enzalutamide) have been successfully used against CRPC; however, resistance eventually develops. A proposed mechanism of acquired resistance to ADT is an adaptive response where CRPC cells switch from being sensitive to the drug target (AR) to a CRPC cell type which is not dependent on the drug target (e.g., neuroendocrine PCa [NEPC]), leading to a reduction or loss of AR [3,4,5]. There are no effective treatments for late-stage advanced CRPC/NEPC resistant to current ADT strategies, suggesting that new approaches are required [3,4,5].

Standard treatments for androgen-sensitive PCa are gonadotropin-releasing hormone (GnRH) agonists such as goserelin and leuprolide, approved in the 1980s by the FDA [6]. These are based on the Nobel Prize discovery of Andrew Schally and others that hormones secreted from the hypothalamus stimulate the release of pituitary hormones (e.g., luteinizing hormone), which regulate androgen synthesis. Since the 1990s, the Schally group has focused their research on the role of growth hormone-releasing hormone (GHRH) in cancer, prompting efforts to develop synthetic antagonists of GHRH that can be used therapeutically [7]. GHRH is a neuropeptide secreted from the hypothalamus that regulates the secretion of growth hormone (GH) from the pituitary, which then stimulates the liver to produce insulin growth factor 1 (IGF1), a potent mitogen for multiple cancers [7,8]. GHRH and its receptor GHRHR (a member of the G protein-coupled receptor [GPCR] family) are also produced in multiple tissues and cancers to modulate cell proliferation and apoptosis, including PCa [7].

The early development of GHRH peptide antagonists improved upon pharmacokinetic properties, target binding, and anti-tumor effects [7]. More recently, the MIA series, especially MIA-602 and -690, have emerged as one of the most promising antagonists by binding to GHRH expressed on tumor cells, blocking GHRHR-mediated signaling pathways, and inhibiting tumor growth [7] (see ref. 67 in [7]). In PCa, GHRH peptide antagonists (including MIA-602) reduced the growth of CRPC xenograft tumors, possibly by decreasing ERK and AKT signaling [9] (see ref. 13 in [9]. However, it is unlikely that the treatment of advanced CRPC/NEPC with GHRH antagonists as single agents will be sufficient for optimal efficacy. There are a few studies that have identified potential drugs that, in combination with GHRH antagonists, can improve efficacy, including DNA-damaging agents (doxorubicin, 5-flourouracil, irinotecan, cisplatin), anti-mitotic docetaxel, and the EGFR inhibitor gefitinib [10]. We searched a series of anti-cancer drugs that, in combination with MIA-602 or -690 GHRH antagonists, increase PCa/CRPC/NEPC cell death.

In this report, we identified phosphatidylinositol 3-kinase (PI3K) isoform inhibitors that, when combined with MIA-602 or -690, increased cell death in all types of PCa, including CRPC and NEPC. PI3K is a family of lipid kinases that increases phosphatidylinositol (3,4,5)-trisphosphate (PIP3) lipids, which are important in mediating signals from receptor tyrosine kinase (RTK) and GPCR to downstream AKT/mTOR signaling pathways [11]. Since the loss of PTEN, a negative regulator of the PI3K pathway, occurs in 40% to 50% of patients with PCa and results in PI3K hyperactivation [12,13], there have been significant efforts to identify PI3K inhibitors that can improve efficacy [11,14,15,16]. Our results showed MIA-602 or -690 + PI3K isoform inhibitors altered multiple signaling pathways including apoptosis, proliferation, PI3Kα/β, AKT, ERK, and AR. The use of MIA-602 and -690 converted into a more clinically relevant acetate salt form had similar results. Overall, the MIA-602 or -690 + PI3K isoform inhibitor combination may improve the therapeutic efficacy in PCa/CRPC/NEPC.

## 2. Materials and Methods

### 2.1. MIA-602 and MIA-690 GHRH Peptide Antagonists

GHRH antagonists MIA-602 (PhAc-Ada^0^, Tyr^1^, d-Arg^2^, 5FPhe^6^, Ala^8^, Har^9^, Tyr(Me)^10^, His^11^, Orn^12^, Abu^15^, His^20^, Orn^21^, Nle^27^, d-Arg^28^, Har^29^-NH2) and MIA-690 (PhAc-Ada^0^-Tyr^1^, d-Arg^2^, Cpa^6^, Ala^8^, Har^9^, 5FPhe^10^, His^11^, Orn^12^, Abu^15^, His^20^, Orn^21^, Nle^27^, d-Arg^28^, Har^29^-NH2) were synthesized and purified as previously described [7] (see ref. 67 in [7]). Changes from the bioactive wild-type human GHRH (1–29) amino acid peptide are provided above. Dried preparations were resuspended in DMSO and small aliquots were stored at −20 °C. The abbreviations are as follows: PhAc, phenylacetate; Ada, 12-aminododecanoyl; 5FPhe, pentafluoro-phenylalanine; Har, homoarginine; Tyr (Me), O-methyltyrosine; Orn, ornithine; Abu, alpha-aminobutanoyl; Nle, norleucine; and Cpa, 4-chloro-phenyalanine. Peptides were eluted from the resin with a solvent containing trifluoroacetic acid (TFA), which is not acceptable for human studies due to its potential subcutaneous toxicity [17]. To remove residual TFA, peptides were passed through a carbonate ion-exchange resin column (VariPure IPE; Agilent, Santa Clara, CA, USA), diluted acetic acid was added, and the samples were lyophilized (referred to as MIA-602Ac [Ac, Acetate salt] and MIA-690Ac). GHRH antagonist activity was confirmed using the GH release assay in rats, as previously described [7] (see ref. 67 in [7]).

### 2.2. Reagents

PI3K inhibitors alpelisib (PI3Kα), AZD8186 (PI3Kβ/δ), and duvelisib (PI3Kδ/γ); AKT1-3 inhibitor AZD5363 (capivasertib); proteasome inhibitor ixazomib (MLN9708); and mTOR inhibitor rapamycin were obtained from APExBIO (Houston, TX, USA); pan-PI3K inhibitor LY294002 and NFκB inhibitor parthenolide were from Sigma-Aldrich (St. Louis, MO, USA); anti-mitotic docetaxel and CDK inhibitor flavopiridol were from Sanofi-Aventis (Bridgewater, NJ, USA); anti-mitotic cabazitaxel was from LC Laboratories (Woburn, MA, USA); Bcl-2 inhibitor ABT-737 was from Abbott Laboratories (Abbott Park, IL, USA); AR antagonist enzalutamide was from Selleckchem (Houston, TX, USA); and Coomassie blue and trypan blue (0.4%) were from Thermo Fisher Scientific (Waltham, MA, USA).

### 2.3. Cell Culture

Human AR+ androgen-sensitive PCa (LNCaP), AR+ CRPC (22Rv1), AR—CRPC (PC3, DU145), AR—NEPC (NCI-H660, LASCPC), and human non-cancer RWPE-1 (prostate epithelial) cells were obtained from the American Type Culture Collection (ATCC) and were used within 6 months of the resuscitation of the original cultures. The molecular characteristics of the PCa/CRPC/NEPC cell lines are summarized in Table 1. The LNCaP, 22Rv1, PC3, and DU145 cells were maintained in RPMI 1640 medium (Thermo Fisher Scientific) and 5% fetal bovine serum (R&D Systems, Minneapolis, MN, USA). The H660 and LASCPC cells were maintained in Advanced DMEM/F12, B27 supplement, Glutamax (Thermo Fisher Scientific), EGF, and bFGF (R&D Systems) [18,19]. RWPE-1 was maintained in Keratinocyte-SFM media (Thermo Fisher Scientific). All cells were grown with 100 U/mL penicillin, 100 μg/mL streptomycin, and 0.25 μg/mL amphotericin (Thermo Fisher Scientific).

### 2.4. Drug Treatments

Cells were cultured in media containing MIA-602, MIA-690 (5 μM, TFA and Ac forms), alpelisib (10 μM), AZD8186 (0.025–10 μM), duvelisib (10 μM), capivasertib (0.025–10 μM), docetaxel (0.25–1 nM), ABT-737 (1 μM), LY294002 (10 μM), cabazitaxel (1 nM), flavopiridol (100 nM), ixazomib (50 nM), rapamycin (0.05 nM), and parthenolide (0.5 μM). In all the experiments, floating and trypsinized attached cells were pooled for further analysis.

### 2.5. Trypan Blue Exclusion Assay to Measure Total Cell Death

Treated and control cells were harvested, resuspended in PBS, diluted 1:1 in 0.4% trypan blue, the dead blue and live non-blue cells were immediately counted using a hemacytometer, and the % of dead blue cells was determined from at least 2–3 independent experiments performed in triplicate.

### 2.6. Cell Proliferation Assay and Determination of Synergy Combination Index (CI)

The CellTiter Aqueous colorimetric method from Promega (Madison, WI, USA) was used to determine the cell proliferation of LNCaP and PC3 cells in media containing MIA-602/690 (TFA, Ac; 1, 2.5, 5 μM), alpelisib (1, 2.5, 5 μM), AZD8186 (5, 25 nM), or the control (0.1% DMSO). Cell proliferation was normalized against DMSO control and the data expressed as a percentage of the control from three independent experiments performed in triplicate. Whether drug interactions were synergistic, additive, or antagonistic was determined using the CalcuSyn Version 2 software program from Biosoft (Cambridge, UK). This program is no longer available from Biosoft. CI ≤ 0.7 was synergistic.

### 2.7. Western Blot Analysis

Cell pellets were resuspended in NP40 cell lysis buffer (1% NP40, 50 mmol/L Tris (pH 8.0), 150 mmol/L NaCl, 2 mmol/L EGTA, 2 mmol/L EDTA, Halt Protease Inhibitor Cocktail, and Pierce Phosphatase Inhibitor [Thermo Fisher Scientific]), lysed by vortex, left on ice for 30 min, centrifuged, and the protein concentrations of the supernatant were determined with the Bio-Rad Laboratories (Hercules, CA, USA) protein assay. After the separation of 25 to 50 μg of protein by SDS-PAGE, the proteins were transferred by electrophoresis to Immobilon-P membrane and incubated in 5% nonfat dry milk, PBS, and 0.25% Tween 20 for 1 h. The following antibodies were used: GHRHR (28692) from Abcam (Waltham, MA, USA); PI3Kα (C73F8), PI3Kβ (C33D4), cl-PARP (9541), phospho (P)-AKT (Ser473; 587F11), AKT (9272), ERK1/2 (9102), and P-ERK1/2 (9101) from Cell Signaling Technology (Danvers, MA, USA); and Mcl-1 (S-19), AR (441), cyclin A (H432), E2F1 (KH59), mouse anti-rabbit IgG-HRP (2357), and m-IgG-Fc BP-HRP (525409) from Santa Cruz Biotechnology (Santa Cruz, CA, USA). Precision Plus Protein Dual Color Standards (Bio-Rad Laboratories) was used to estimate the molecular weights in kDa. Markers were used to cut the blots into horizontal strips so high, medium, and low molecular weight targets could be analyzed separately with the appropriate antibodies. In some cases, after analysis, the strips were pretreated with methanol for 1 min, washed, treated with Ponceau S Staining Solution (Thermo Fisher Scientific) for 15 min to strip the antibody signal, and analyzed with a different antibody. After immunodetection, our preference for loading controls was the staining of total proteins transferred to the membrane with Coomassie blue, because drug treatments often affect the levels of typical housekeeping proteins such as actin or tubulin. Blot images were cropped for the clarity of the presentation. Quantification of protein bands from images (Chem Doc MP Imaging System, Bio-Rad Laboratories) was performed using the UN-SCAN-IT digitizing software version 5.1 from Silk Scientific (Provo, UT, USA) (normalized to the protein signal from Coomassie blue staining). Measurements (pixels) from the specific protein signal were divided by the Coomassie blue stain protein signal and the final fold changes were determined by dividing this over the control cells (=1).

### 2.8. Statistics

Statistical differences between drug-treated and control cells were determined by two-tailed Student’s *t*-test (unequal variance) from 2 to 3 independent experiments performed in duplicate or triplicate with *p* < 0.05 considered significant. GraphPad Prism software (UN-SCAN-IT version 5.1) (Boston, MA, USA) was also used to calculate statistical differences using one way ANOVA followed by Dunnett’s or Šidák’s multiple comparisons test. The experimental data were presented as mean ± standard deviation.

## 3. Results

### 3.1. Database Analysis of GHRHR and GHRH mRNA Expression in PCa

Since GHRHR and its ligand GHRH are expressed together in PCa specimens and likely stimulate proliferation [9] (see ref. 14 in [9]), we analyzed their mRNA expression using known databases. In the Oncomine cancer microarray database (shut down 17 January 2022), GHRHR mRNA had a 1.65-fold higher expression in PCa compared to normal tissue (Figure 1A; LaTulippe Prostate Statistics). Interestingly, GHRHR expression appeared to be higher in metastatic compared to primary PCa, and the location of higher GHRHR metastasis appeared to be in soft tissue compared to bone (Figure 1A). This observation requires further rigorous investigation to confirm or refute it. In the Gene Expression Profiling Interactive Analysis database (gepia.cancer-pku.cn), GHRHR mRNA also appeared to be slightly higher in PCa compared to normal tissue (Figure 1B). However, there was no statistical difference in low vs. high GHRHR expression in relation to disease-free survival (Figure 1B). In the OncoDB database (oncodb.org), GHRH expression was slightly higher in PCa compared to normal tissue (*p* = 0.037) (Figure 1C). Overall, the results from the databases supported a higher expression of GHRHR and GHRH mRNA in PCa, although the differences were small. This weak correlation provided some justification to investigate combinations that enhance GHRH antagonists as a therapeutic strategy.

### 3.2. Searching for a Drug Combination with MIA-602 and MIA-690 GHRH Antagonist Peptides to Increase Cell Death in PCa/CRPC/NEPC

Since GHRHR is expressed in PCa, the addition of GHRH antagonists should block the key signals important for proliferation and survival. Results indicated different sensitivities of PCa/CRPC/NEPC cells to MIA-602 and -690 (Figure 2). AR expressing LNCaP and 22Rv1 had greater cell death compared to CRPC PC3 and DU145, whereas NEPC cells H660 and LASCPC appeared to have some sensitivity to MIA-602 and -690. However, it is likely that the treatment of PCa/CRPC/NEPC with GHRH antagonists as single agents would not be sufficient for optimal efficacy.

We searched a series of anti-cancer drugs that, in combination with GHRH antagonists, can increase PCa/CRPC/NEPC cell death. The combination of the anti-mitotic docetaxel + MIA-602 or -690 increased cell death in LNCaP, 22Rv1, and PC3, supporting previous results (Figure 3A) [10](see ref. 13 in [10]). However, the second-generation anti-mitotic cabazitaxel + MIA-602 or -690 did not enhance cell death in LNCaP (Appendix A). Similarly, the combination of the Bcl-2 inhibitor ABT-737, the CDK inhibitor flavopiridol, and the proteasome inhibitor ixazomib + MIA-602 or -690 did not consistently enhance cell death in LNCaP, 22Rv1, and PC3 (Figure 3B and Appendix A). However, the combination of the pan-PI3K inhibitor LY294002 + MIA-602 or -690 consistently increased cell death in LNCaP, 22Rv1, and PC3 (Figure 3C).

### 3.3. PI3Kα or β Isoform Inhibitors + MIA-602 or -690 Increases Cell Death in PCa/CRPC/NEPC

Since LY294002 is a laboratory tool and cannot be used clinically, we identified PI3K isoform inhibitors that are FDA approved or in clinical studies that can be combined with MIA-602 or -690 [11,14,15,16]. In LNCaP and DU145, the PI3Kβ inhibitor AZD8186 (PI3Kβi) but not the PI3Kα inhibitor alpelisib (PI3Kαi) + MIA-602 or -690 was more effective at increasing cell death (Figure 4A,B). In contrast, PI3Kαi + MIA-602 or -690 was more effective in 22Rv1 and PC3 at increasing cell death (Figure 4A,B). In NEPC cells, PI3Kαi was more effective in H660 whereas both PI3Kαi and PI3Kβi increased MIA-602 or -690 cell death in LASCPC (Figure 4C). Appendix A showed similar *p* values when some of the data from Figure 2, Figure 3C and Figure 4A,B were recalculated using ANOVA statistical analysis. The PI3Kδ/γ inhibitor duvelisib did not increase cell death + MIA-602 or -690 in DU145, H660, and LASCPC (Appendix A).

The AKT1-3 inhibitor AZD5363 (capivasertib) or the mTOR inhibitor rapamycin (both are downstream mediators of PI3K signaling [11,14,15,16]) had little or no effect on increasing cell death with MIA-602 or -690 in DU145, H660, and LASCPC (Appendix A). Similarly, the NFκB inhibitor parthenolide did not increase cell death with MIA-602 or -690 in LASCPC (Appendix A). Overall, we identified isoform inhibitors of PI3Kα or β as being the best combination with MIA-602 or -690 to increase cell death in PCa/CRPC/NEPC cells.

### 3.4. MIA-602/690 Alone and + PI3K Inhibitors Alters Multiple Signaling Pathways and AR Expression

The molecular characteristics of the PCa/CRPC/NEPC cell lines used in this study are summarized in Table 1. The negative regulator of PI3K signaling, PTEN, is mutated/deleted in LNCaP, PC3, and H660 [20,21], resulting in constitutively active (ca) AKT signaling. When PTEN activity appears normal, as in 22Rv1 and DU145 [20,22], basal AKT signaling is low. The LASCPC NEPC cells were derived from the overexpression of N-Myc and AKT myristoylated (ca AKT signaling) [18].

Western blot analysis indicated MIA-602 + PI3Kαi decreased PI3Kα and PI3Kβ whereas MIA-602 alone decreased PI3Kα in 22Rv1 and PC3 (Figure 5A). MIA-602 + PI3Kαi decreased AKT in 22Rv1 and PC3 and increased P-AKT in PC3 (24 h, not at 72 h). Changes in ERK (RTKs often crosstalked with GHRHR signaling [23]) were noted: (1) MIA-602 decreased P-ERK in 22Rv1 at 72 h (counters increased P-ERK with PI3Kαi); (2) MIA-602 alone and MIA-602 + PI3Kαi decreased the total ERK in 22Rv1 (72 h); and (3) there was a switch from P-ERK1 to P-ERK2 with MIA-602 and MIA-602 + PI3Kαi in PC3 (24, 72 h). No clear differences were noted in GHRHR and cl-PARP.

In LNCaP, MIA-602/690 alone and MIA-602/690 + PI3Kβi decreased Mcl-1L (anti-apoptosis) and increased Mcl-1S (pro-apoptosis); however, no clear differences in the apoptosis marker cl-PARP were noted (Figure 5B). PI3Kβi alone and MIA-602/690 + PI3Kβi decreased proliferation markers E2F1 and cyclin A. The decrease in PI3Kα and PI3Kβ was stronger in MIA-690 + PI3Kβi. Decreased P/T-AKT was stronger with PI3Kβi alone and MIA-602/690 + PI3Kβi. Interestingly, AR was strongly decreased with MIA-602/690 alone. The treatment of LNCaP with MIA-602/690 alone over time (4–72 h) resulted in similar effects on Mcl-1L, Mcl-1S, and AR (Figure 5C). There was a rapid and sustained decrease in PI3Kα and PI3Kβ (4 h) and decreased P-AKT occurred by 24 h. No clear differences were noted in GHRHR.

A comparison of PI3Kα and PI3Kβ protein levels in non-treated cells revealed that (1) PI3Kα was highest in PC3 and 22Rv1 and (2) PI3Kβ was highest in LNCaP (Appendix A). Although correlative, these results may provide information about why PC3 and 22Rv1 respond better to PI3Kαi and LNCaP responds better to PI3Kβi. Overall, the results suggested that MIA-602/690 alone and in combination with PI3Kα or β isoform inhibitors altered multiple signaling pathways including apoptosis, proliferation, PI3Kα/β, AKT, ERK, and AR.

### 3.5. Testing MIA-602 and -690 Acetate Salt (Ac) Forms for Future Clinical Applications

To prepare for clinical applications, MIA-602 and -690 were converted from TFA to an Ac salt form to reduce potential toxicity from subcutaneous injection (see Methods for description). A comparison revealed that MIA-602Ac and -690Ac were better at increasing cell death in 22Rv1 compared to MIA-602/690TFA (Appendix A). In LNCaP and 22Rv1, MIA-602Ac and -690Ac increased cell death when combined with PI3Kβi or PI3Kαi, similar to results obtained with MIA-602/690 in the TFA form (Figure 6A). Cell proliferation assays revealed that MIA-602/690 alone (Ac or TFA form) had little or modest effects in LNCaP and PC3 (Appendix A). Calcusyn analysis of cell proliferation assays revealed that MIA-602Ac and -690Ac + PI3Kαi or PI3Kβi synergistically inhibited LNCaP and PC3 proliferation (CI = 0.11–0.39, strong synergy) (Appendix A). Similar results were obtained with MIA-602/690TFA + PI3Kαi or PI3Kβi.

Western blot analysis in LNCaP indicated that MIA-602Ac/690Ac + PI3Kβi increased cl-PARP better than either one alone (24, 48 h), which differed from MIA-602/690 in TFA form (Figure 6B). MIA-602Ac/690Ac’s effects on Mcl-1L and Mcl-1S were similar but not as strong as with TFA forms. There were some similarities with MIA-602Ac/690Ac’s effects on E2F1, cyclin A (proliferation), and AR. Stronger effects of MIA-690Ac on reducing GHRHR (24 h) were noted and ERK was increased by PI3Kβi. Similar results were obtained in 22Rv1 with the MIA-602Ac/690Ac + PI3Kαi combination, but with MIA-690Ac demonstrating stronger effects (Figure 6C). Overall, these results indicated that the process of converting MIA-602 and -690 from the TFA to the Ac salt form did not affect its activity. In addition, MIA-602Ac also had similar anti-tumor activity in an H460 non-small cell lung cancer xenograft. Thus, the application of MIA602Ac and -690Ac for future clinical studies can proceed.

### 3.6. AR Antagonist Enzalutamide + MIA-602 or -690 Has No Effect

Because of the unexpected result that MIA-602 and -690 greatly reduced AR in LNCaP and 22Rv1 (Figure 5B,C and Figure 6B,C), we investigated whether adding the AR antagonist enzalutamide could improve efficacy. Our reasoning was that if AR levels were reduced, the addition of enzalutamide would be more effective due to the presence of less AR targets. This rationale was previously supported by using a deubiquitinase inhibitor (betulinic acid) to reduce AR and by adding enzalutamide to enhance apoptotic cell death in LNCaP [24]. However, the current results indicated that enzalutamide did not enhance cell death mediated by MIA-602/690 in LNCaP (Appendix A). The likely reason is that enzalutamide increased the AKT signaling pathway, which is known to negatively regulate the AR signaling pathway [25,26]. Therefore, we predict MIA-602 or -690 + enzalutamide will not prove to be an effective combination.

## 4. Discussion

There is clear evidence in cell line and mouse models of PCa that targeting GHRH with peptide antagonists provides some anti-cancer efficacy [9]. However, it is unlikely that GHRH antagonists such as MIA-602 or -690 will provide optimal therapeutic benefit as single agents, especially for advanced CRPC/NEPC. Our data identified clinically relevant PI3K isoform inhibitors (alpelisib [PI3Kαi]; AZD8186 [PI3Kβi]) + MIA-602 or -690 as a novel strategy for the treatment of PCa/CRPC/NEPC. We suggest that these combinations alter multiple signaling pathways including apoptosis, proliferation, PI3Kα/β, AKT, ERK, and AR to amplify the effects on cell death.

MIA-602 and -690 are potent inhibitors of GHRHR signaling in multiple cancers but have weaker effects on GH/IGF-1 release, suggesting an important autocrine mechanism [7]. Our results revealed some pro-cell death effects in PCa, CRPC, and NEPC. There is a suggestion that since MIA-690 blocks the GHRH-mediated increase in NE differentiation of LNCaP, its use in NEPC therapy should be considered [27]. Overall, the effects of MIA-602/690 as single agents were minor, prompting us to search for a drug combination that could increase cell death.

Previous data showed the effects of GHRH antagonists on Twist/N-cadherin, NFκB, JAK2/STAT3, AKT, and ERK, suggesting that multiple signaling pathways are altered [7,23,28]. More recently, data have suggested that the splice variant of GHRHR (SV1) increases the intracellular recruitment of β-arrestins to enhance cell proliferation via ERK signaling, whereas GHRHR predominantly activates guanine nucleotide binding (G) proteins to increase cAMP signaling [29]. MIA-602/690 have been shown to decrease cAMP signaling [23], but whether alterations in β-arrestin coupling are altered is not known. Overall, no clear consensus mechanism has been elucidated. Our data in PCa/CRPC were supportive of MIA-602/690 altering multiple signaling pathways. New data not previously reported were the effects on switching anti-apoptotic Mcl-1L to pro-apoptotic Mcl-1S [30] and the decreased AR, E2F1, cyclin A, and PI3Kα/β. The mechanisms of how MIA-602/690 as single agents can alter individual proteins in multiple signaling pathways requires further investigation.

Our data revealed that adding PI3K isoform inhibitors to MIA-602/690 increased cell death in all types of PCa, including CRPC and NEPC. No other drug combination provided as consistent an increase in cell death as the combination with PI3K inhibitors. PIK3CA is one of the most frequently mutated oncogenes in cancer and therefore provides a strong candidate for drug development and targeting [31]. In fact, PIK3CA mutation/amplification correlates with poor survival and occurs in ~60% of PCa patients [32]. There is a suggestion in PTEN-deficient PCa that PI3Kα activity is suppressed and PI3Kβ drives PI3K signaling [33]. However, our data suggested that sensitivity to PI3K isoform inhibitors + MIA-602/690 may depend on PI3Kα or PI3Kβ protein levels rather than PTEN status [34]).

In contrast to PI3K, the inhibition of AKT, mTOR, or NFκB did not increase cell death in combination with MIA-602/690. One possibility is that the phosphoinositide-dependent kinase-1 (PDK1) pathway is more prominent in GHRHR signaling. PI3K increases PIP3 to serve as a membrane scaffold for PDK1, a key upstream regulator in the AKT/mTOR pathway. In addition to partially activating AKT via phosphorylation, PDK1 also phosphorylates several other kinases including protein kinase C, serum/glucocorticoid-regulated kinase, polo-like kinase, ribosomal S6 kinase, and others [35]. In PTEN-intact cancer cells with low AKT activity, PI3K mainly utilizes AKT-independent signals via PDK1 and serum/glucocorticoid-regulated kinase-3 (SGK3) [36]. In PTEN-mutant cancer cells with high AKT activity, PI3K utilizes PDK1 and AKT signals. Presumably, PI3K isoform inhibitors will reduce PIP3 and decrease PDK1/AKT membrane localization. There is evidence that ribosomal protein S6 kinase beta 1 phosphorylates PDK1 to reduce interactions with PIP3 and decrease AKT activation [37]. PDK1 regulates cell survival in CRPC cells through the activation of SGK3 signaling, suggesting that they are potential drug targets [38]. However, inhibitors of PDK1 or SGK3 have yet to be developed for clinical use.

The identification and development of PI3K isoform inhibitors has been prompted by the clinical observation that pan-PI3K inhibitors have intolerable toxicity [14]. The PI3Kα inhibitor alpelisib is approved by the FDA for the treatment of PIK3CA-mutated advanced or metastatic breast cancer, with manageable toxicity [39]. The PI3Kβ inhibitor AZD8186 has completed a Phase I clinical trial in solid tumors with PTEN deficiency or PI3Kβ mutation, but with limited efficacy [40]. In PCa, Phase I results for alpelisib and AZD8186 are limited or have been disappointing. The effectiveness of PI3K isoform inhibitors in cancer therapy is challenging due to the activation of adaptive mechanisms and crosstalk with other signaling pathways [41]. Therefore, a combination with other drugs to improve efficacy is required. Given that MIA-602/690 can alter multiple signaling pathways, a combination with PI3K isoform inhibitors may weaken the activation of adaptive mechanisms and improve efficacy. Further investigations are required to determine if this is correct.

To prepare for pre-Investigational New Drug (IND) studies, MIA-602/690 were converted into a more clinically acceptable Ac salt form (see Methods). Our results revealed that MIA-602Ac and -690Ac remained potent with regard to their effects on multiple signaling pathways (apoptosis, proliferation) and in combination with PI3K isoform inhibitors (synergistic, CI < 0.4). Overall, our results suggested that MIA-602/690 alone and in combination with PI3Kα or β isoform inhibitors altered multiple signaling pathways including apoptosis, proliferation, PI3Kα/β, AKT, ERK, and AR.

The limitations of this study are (1) a lack of testing of the MIA-602/690 + PI3K isoform inhibitor combination in a mouse model of CRPC/NEPC and (2) a lack of identification of a specific mechanism for why PI3K isoform inhibitors enhance MIA-602/690 cell death. Future studies will (1) develop shRNA-knockdown or CRISPR-deletion PI3K isoforms to validate whether PI3Kα or PI3Kβ are the key players in synergy with MIA-602/690; (2) determine if PDK1 has an important role in GHRHR signaling; (3) use CRPC/NEPC xenograft models (e.g., 22Rv1, PC3, H660, or patient-derived xenografts) to confirm the efficacy of the combination in a preclinical setting; (4) determine whether MIA-602/690 (less toxic) can lower the dose of PI3K isoform inhibitors in order to reduce toxicity without affecting anti-tumor efficacy; and (5) investigate the mechanisms of potential resistance mediated by MIA602/690 + PI3K isoform inhibitors.

## 5. Conclusions

Discovering new strategies for the treatment of advanced CRPC/NEPC is an urgent topic. Our MIA-602/690 + PI3K isoform inhibitor combination strategy may improve upon the efficacy of each drug used alone, especially in advanced CRPC/NEPC. We consider that due to the reciprocal negative feedback regulation between PI3K/AKT activity and AR signaling [25,26], the MIA-602/690-mediated decrease in AR in LNCaP and 22Rv1 may increase PI3K activity. Therefore, the addition of PI3K inhibitors may counter the negative feedback effect of AR reduction. In addition, PI3K inhibitors should further block the RTK/ERK signaling pathway (shared PI3K family members) in AR-negative CRPC/NEPC above the RTK/ERK inhibition resulting from GHRH antagonists [23] (Figure 7). Our future goal is to translate MIA-602Ac or -690Ac to the clinic and eventually investigate whether the combination with PI3K isoform inhibitors will provide improved therapeutic benefits, especially for advanced CRPC/NEPC.

## Figures and Tables

**Figure 1 cancers-17-01643-f001:**
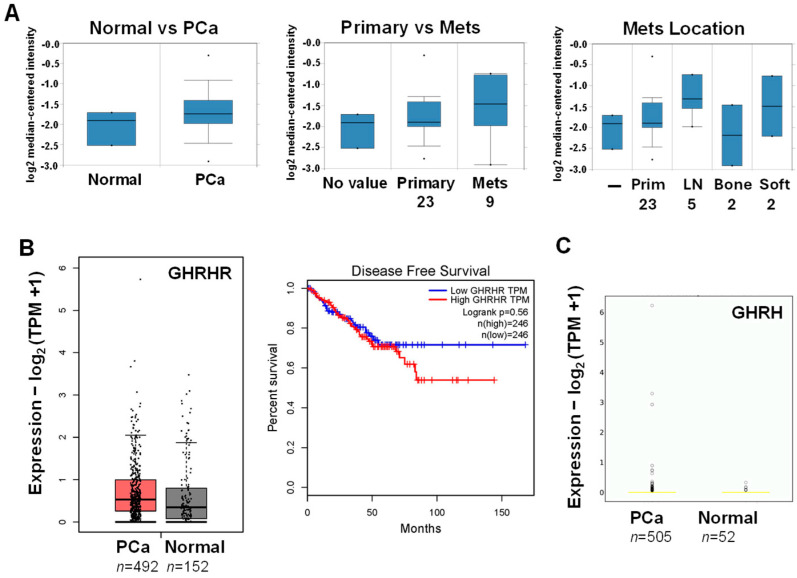
Higher expression of GHRHR and GHRH mRNA in PCa compared to normal tissue: a database analysis. (**A**) The Oncomine database search indicated a 1.65-fold higher expression for GHRHR mRNA in PCa vs. normal tissue (LaTulippe Prostate Statistics; *p* = 0.008). GHRHR mRNA was higher in metastatic (mets; *n* = 9) vs. primary (*n* = 23) PCa. Higher GHRHR mRNA metastasis was localized more in soft tissue (LN, lymph node; *n* = 5) vs. bone (*n* = 2). (**B**) The GEPIA database search indicated a higher expression of GHRHR mRNA in PCa (*n* = 492) vs. normal tissue (*n* = 152). High vs. low GHRHR expression was not correlated with disease-free survival (*n* = 246 for each). (**C**) The OncoDB database search indicated the GHRH mRNA expression was higher in PCa (*n* = 505) vs. normal tissue (*n* = 52) (*p* = 0.037).

**Figure 2 cancers-17-01643-f002:**
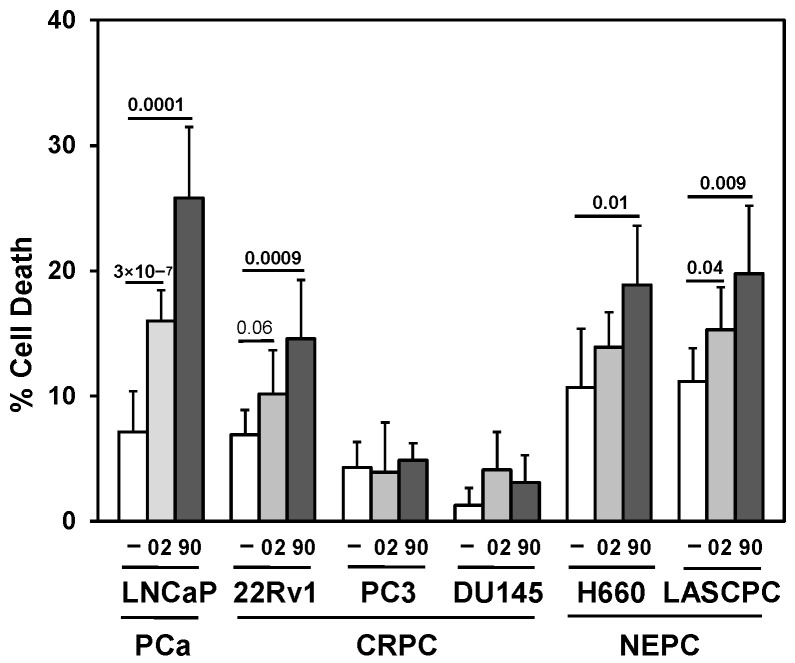
MIA-602 and -690 GHRH antagonists increase cell death in PCa/CRPC/NEPC cells. Trypan blue exclusion assay showed significantly higher cell death (72 h) in MIA-602 (02) and -690 (90) (5 μM)-treated LNCaP (PCa), 22Rv1 (CRPC), and H660 LASCPC (NEPC) cells compared to control (—)-treated cells. There was no increased cell death in PC3 or DU145 (CRPC). *P* values are shown above the bars (bold numbers indicate significance).

**Figure 3 cancers-17-01643-f003:**
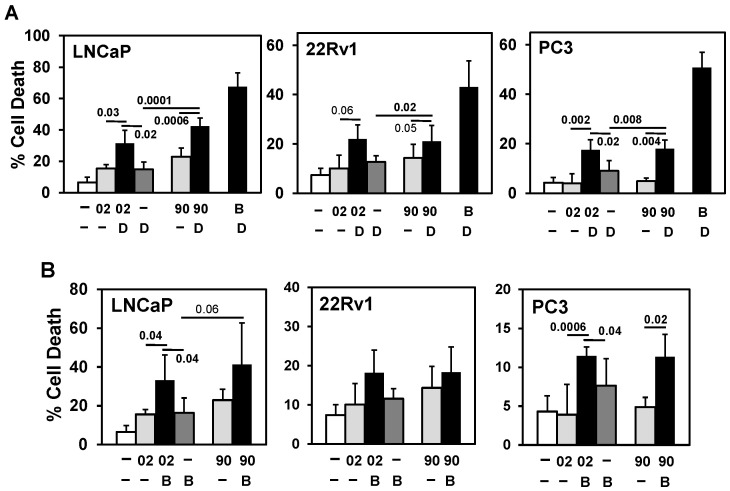
Searching for a drug combination with MIA-602 and -690 GHRH antagonist peptides that will increase cell death in PCa/CRPC/NEPC. (**A**) Trypan blue exclusion assay shows that anti-mitotic docetaxel (D; 0.25 nM LNCaP, 22Rv1; 1 nM PC3) + MIA-602 (02) or -690 (90) (5 μM) increases cell death in LNCaP, 22Rv1, and PC3 cells compared to D, 02/90, and control cells. Positive control is D + Bcl-2 (B) inhibitor ABT-737 (1 μM). (**B**) Bcl-2 (B) inhibitor ABT-737 (1 μM) + MIA-602 or -690 partially increases cell death in LNCaP and PC3 but not in 22Rv1 cells. (**C**) Pan-PI3K inhibitor LY294002 (LY, 10 μM) + MIA-602 or -690 significantly increases cell death in LNCaP, 22Rv1, and PC3 compared to LY, 02/90, and control cells. *P* values are shown near the bars (bold numbers indicate significance).

**Figure 4 cancers-17-01643-f004:**
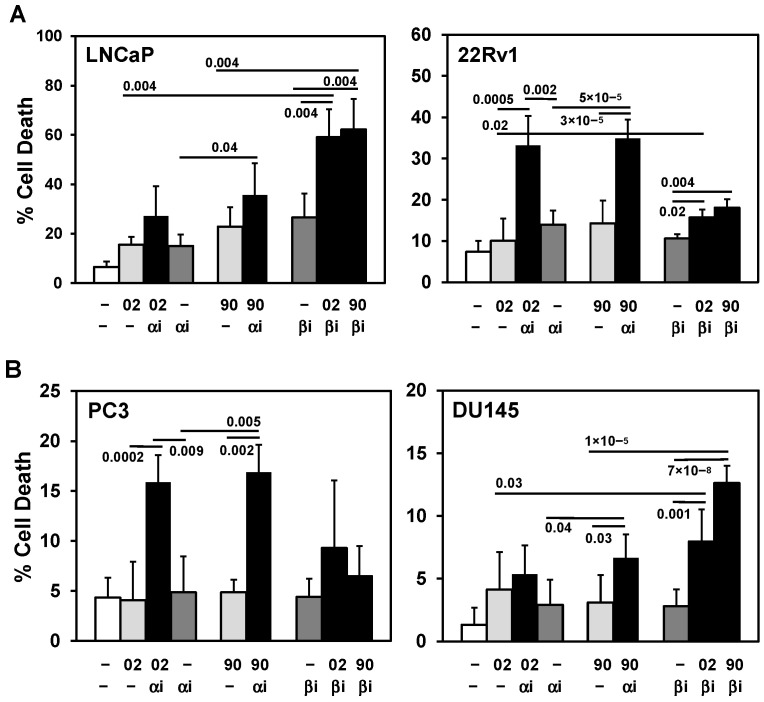
PI3Kα or β isoform inhibitors + MIA-602 or -690 increase cell death in PCa/CRPC/NEPC cells. (**A**) Trypan blue exclusion assay shows PI3Kαi (αi, 10 μM) + MIA-602 (02) or -690 (90) (5 μM) significantly increase cell death in 22Rv1 compared to αi, 02/90, and control cells. In LNCaP, PI3Kαi does not increase MIA-602 or -690 cell death. PI3Kβi (βi, 2.5 μM) + MIA-602 or -690 significantly increase cell death in LNCaP compared to βi, 02/90, and control cells, but only slightly increase cell death in 22Rv1 (βi, 10 μM). (**B**) PI3Kαi + MIA-602 or -690 significantly increase cell death in PC3 but partially in DU145 (90 + αi). PI3Kβi (10 μM) significantly increases MIA-602 or -690 cell death in DU145 but not in PC3. (**C**) PI3Kαi + MIA-602 or -690 significantly increase cell death in H660 and LASCPC whereas PI3Kβi (10 μM) + MIA-602 or -690 slightly increase cell death in LASCPC but not in H660. *P* values are shown near the bars (bold numbers indicate significance).

**Figure 5 cancers-17-01643-f005:**
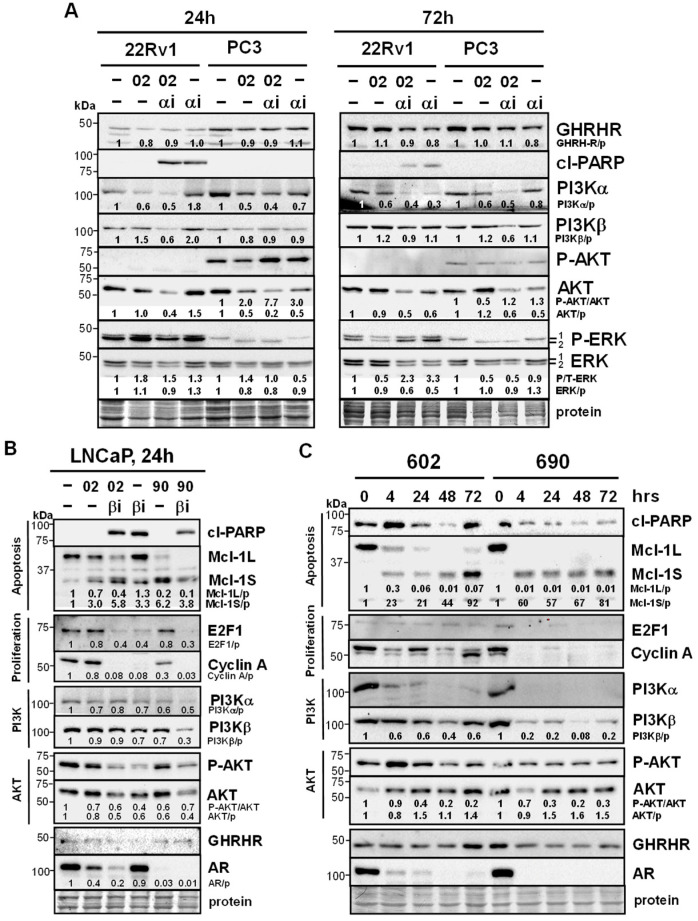
MIA-602/690 alone and + PI3K isoform inhibitors alter multiple signaling pathways and AR expression. (**A**) Western blot analysis showed that MIA-602 (02, 5 μM) + PI3Kαi (αi, 10 μM) decreased PI3Kα, PI3Kβ, and AKT in 22Rv1 and PC3 (P-AKT increased in PC3 24 h); MIA-602 alone decreased PI3Kα in 22Rv1. MIA-602 decreased P-ERK in 22Rv1 (72 h) (PI3Kαi increased P-ERK). MIA-602 alone and MIA-602 + PI3Kαi decreased the total ERK in 22Rv1 (72 h). In PC3, there was a switch from P-ERK1 to P-ERK2 with MIA-602 and MIA-602 + PI3Kαi (24, 72 h). No clear differences were noted in GHRHR and cl-PARP. (**B**) In LNCaP, MIA-602/690 alone and MIA-602/690 + PI3Kβi (βi, 2.5 μM) (24 h) decreased Mcl-1L (anti-apoptosis) and increased Mcl-1S (pro-apoptosis). No clear difference in the apoptosis marker cl-PARP was noted. PI3Kβi alone and MIA-602/690 + PI3Kβi decreased proliferation markers E2F1 and cyclin A. MIA-690 + PI3Kβi decreased PI3Kα, PI3Kβ, and P/T-AKT. AR was strongly decreased with MIA-602/690 alone. (**C**) In LNCaP, treatment with MIA-602/690 over time (4–72 h) decreased Mcl-1L, PI3Kα, PI3Kβ, P-AKT, and AR. No clear differences were noted in GHRHR. Protein refers to the Coomassie blue stain of blots after all immunological analysis was completed. Quantification values (divided by protein [p]) for PI3Kα, PI3Kβ, AKT, ERK, Mcl-1L, Mcl-1S, E2F1, cyclin A, and AR are shown below specific bands with control = 1. The ratio of P/T (total) AKT and ERK values are also shown.

**Figure 6 cancers-17-01643-f006:**
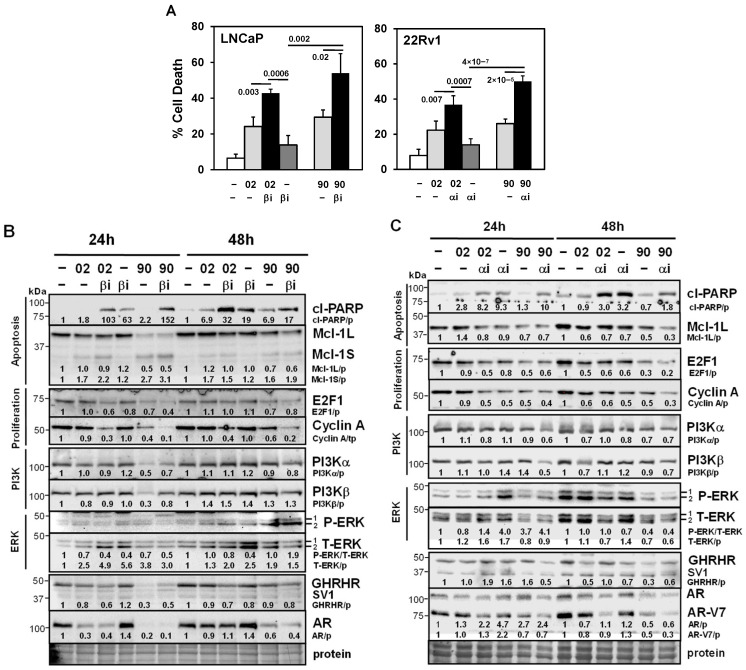
MIA-602Ac and -690Ac are clinically relevant forms with similar effects on signaling pathways. (**A**) Trypan blue exclusion assay showed that MIA-602Ac (02) or -690Ac (90) (5 μM) + PI3Kβi (βi, 25 nM) (LNCaP) or PI3Kαi (αi, 10 μM) (22Rv1) significantly increased cell death compared to βi/αi, 02/90, and control cells. *p* values are shown near the bars. (**B**) Western blot analysis in LNCaP showed that MIA-602Ac or -690Ac (5 μM) + PI3Kβi (25 nM) increased cl-PARP better than either one alone (24, 48 h). MIA-602Ac/690Ac decreased Mcl-1L (apoptosis), E2F1, cyclin A (proliferation), GHRHR (stronger with 90 at 24 h), and AR, and increased Mcl-1S. ERK was increased by PI3Kβi. (**C**) In 22Rv1, Western blot results showed that MIA-602Ac/690Ac + PI3Kαi (10 μM) had similar changes compared to LNCaP, with MIA-690Ac demonstrating stronger effects. Protein refers to the Coomassie blue staining of blots after all immunological analysis was completed.

**Figure 7 cancers-17-01643-f007:**
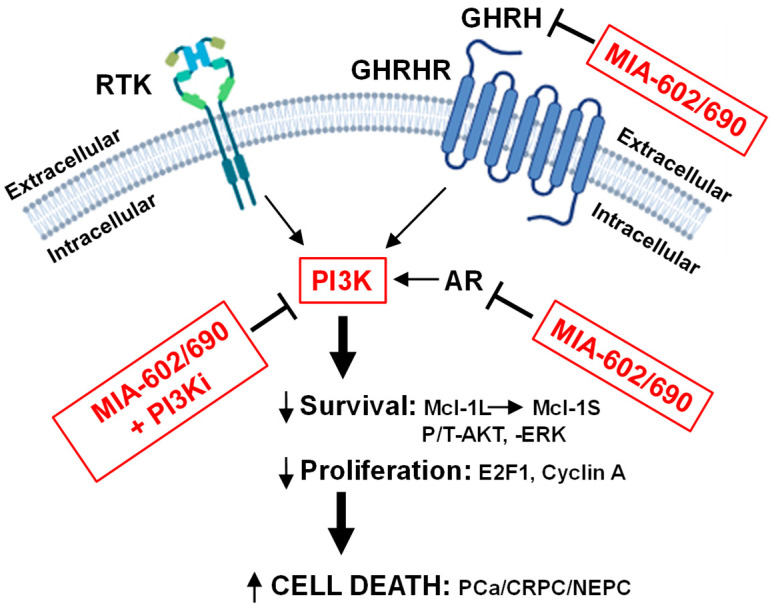
Schematic summary of how MIA-602/690 alone and + PI3K isoform inhibitors are hypothesized to affect multiple signaling pathways in PCa/CRPC/NEPC. MIA-602/690 antagonize extracellular GHRH to block GHRHR intracellular signaling. A likely consequence of the MIA-602/690 decrease in intracellular AR is an increase in PI3K activity, which is countered by adding PI3Kα or PI3Kβ isoform inhibitors. In CRPC/NEPC cells that are AR-negative, MIA-602/690 + PI3Ki should further reduce signaling from RTK (decreased PI3K and altered ERK). The downstream effects are a reduction in survival (anti-apoptotic Mcl-1L switch to pro-apoptotic Mcl-1S; decreased P/T-AKT/ERK) and proliferation (decreased E2F1, cyclin A), and increased cell death in all types of PCa, including CRPC/NEPC.

**Table 1 cancers-17-01643-t001:** Characteristics of prostate cancer cell lines used in the current study.

Cell Line	Type	AR	PTEN	AKT	p53	Additional
LNCaP	AS PCa	+	mut/−	ca	+/+	ARmut T877A
22Rv1	CRPC	+	+/+	wt	+/+	(1) AR-V7 (splice variant—LBD) (2) PIK3CA mut
PC3	CRPC	−	−/−	ca	−/−	
DU145	CRPC	−	+/−	wt	dn/oe	Bax null
H660	NEPC	−	−/−	ca	mut	
LASCPC	NEPC	−		ca		N-myc/AKTmyr oe

AS, androgen-sensitive; mut, mutant; ca, constitutively active; wt, wild-type; LBD, ligand-binding domain; dn, dominant negative; oe, overexpressed; myr, myristoylated.

## Data Availability

All data generated or analyzed during this study are included in this published article and the Appendix A. Upon written or e-mail request, any resources or data will be made freely available.

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
