# Peer review of "Growth Hormone-Releasing Hormone (GHRH) Antagonist Peptides Combined with PI3K Isoform Inhibitors Enhance Cell Death in Prostate Cancer"

_cancers, 2025, doi:10.3390/cancers17101643_

Round 1
Reviewer 1 Report
Comments and Suggestions for Authors
The report I revised is well organized, and the experiments are appropriate. The purpose of this work is clearly understood, and the use of the therapeutic strategy with GHRH and PI3K inhibitors can be an excellent approach. The only thing that was not addressed is the potential resistance mediated by GHRH and PI3K inhibitors, but this could be developed further (add the concept for discussion).
Author Response
The report I revised is well organized, and the experiments are appropriate. The purpose of this work is clearly understood, and the use of the therapeutic strategy with GHRH and PI3K inhibitors can be an excellent approach. The only thing that was not addressed is the potential resistance mediated by GHRH and PI3K inhibitors, but this could be developed further (add the concept for discussion).
Thank you for your review. We expanded the “Future studies” section in the discussion to address this possibility.
Reviewer 2 Report
Comments and Suggestions for Authors
The manuscript titled “Growth Hormone-Releasing Hormone (GHRH) Antagonist Peptides Combined with PI3K Isoform Inhibitors Enhances Cell Death in Prostate Cancer” provides insights into combining GHRH antagonists (MIA-602/690) with PI3K inhibitors enhances prostate cancer cell death by counteracting feedback activation of PI3K signaling, affecting multiple pathways, and potentially improving therapeutic efficacy. This is an interesting study but require attention to following comments:
- How does the inhibition of PI3K isoforms (PI3Ka or PI3Kb) specifically enhance the pro-apoptotic effects of MIA-602/690 in prostate cancer cells? Have authors checked any specific apoptosis marker??
- Please specify what does “protein” mean in immunoblot representation and how fold changes were analyzed.
- Have authors checked internal control such as actin or GAPDH in immunoblot studies?
- The paragraph lengths in some sections, such as the discussion, are inconsistent—either too long or too short. Authors are advised to maintain uniformity for better readability.
- Authors need to specify if the combination of MIA-602/690 and PI3K inhibitors overcome resistance mechanisms in prostate cancer with PTEN loss or PIK3CA mutations more effectively than current therapies.
- Authors need to discuss the potential off-target effects or toxicity concerns of combining GHRH antagonists with PI3K inhibitors in preclinical and clinical settings
- Discussion section needs to avoid repetition of result section.
- A deeper discussion of study limitations, challenges, and future research directions, along with a more detailed introduction and discussion, would enhance the manuscript's completeness. To address this deficiency and align with the relevant findings presented in the manuscript, the authors are encouraged to integrate pertinent literature (please refer to PMID: 39505776, 35895804, 38965223). This literature notably delves into the modifications of critical signaling pathways, elucidating alterations in growth in prostate cancer including GHRH. By integrating these studies into the manuscript, the authors can substantially strengthen the basis for their compelling and comprehensive concept of GHRH/PI3K axis in regulation of prostate cancer.
- The manuscript requires a significant attention to improve punctuations, grammar and the readability specifically in discussion section.
Author Response
- How does the inhibition of PI3K isoforms (PI3Ka or PI3Kb) specifically enhance the pro-apoptotic effects of MIA-602/690 in prostate cancer cells? Have authors checked any specific apoptosis marker??
The apoptosis markers we investigated were cl-PARP and Mcl-1. MIA-602 (TFA form) + PI3Kai did not show a difference in cl-PARP in 22Rv1 and PC3 (Fig. 5). However, later studies using MIA-602 and -690 (Ac form) + PI3Kbi increased cl-PARP in LNCaP, suggesting enhancement of apoptosis (Fig. 6B). MIA-602 and -690 (TFA and Ac) alone decreased Mcl-1L (anti-apoptotic) and increased Mcl-1S (pro-apoptotic) in LNCaP (Fig. 5B, C, and 6B). In combination with PI3Kbi, there was a greater decrease in Mcl-1L in LNCaP (Fig. 5B) and a small increase in Mcl-1S (Fig. 6B). In 22Rv1, MIA-690Ac + PI3Kai decreased Mcl-1L at 48h (Fig. 6C). The specific mechanisms on how PI3K isoform inhibitors enhance MIA-602/690 apoptosis effects requires further investigation.
- Please specify what does “protein” mean in immunoblot representation and how fold changes were analyzed.
To clarify, we add the sentence “Protein refers to Coomassie blue stain of the blot after all immunological analysis was completed” in the figure legends for the Western blots presented in Figs. 5B, 5C, 6B, 6C, and Supplementary Figs. S6, S9. In section 2.7, we stated “Quantification of protein bands from images (Chem Doc MP Imaging System, Bio-Rad Laboratories) was done using the UN-SCAN-IT digitizing software from Silk Scientific (Provo, UT, USA) (normalized to protein signal from Coomassie blue stain).” We add the sentence “Measurements (pixels) from the specific protein signal were divided by the Coomassie blue stain protein signal and the final fold changes were determined by dividing over the control cells (=1).”
- Have authors checked internal control such as actin or GAPDH in immunoblot studies?
In section 2.7, we stated “After immunodetection, our preference for loading controls was staining of total proteins transferred to the membrane with Coomassie blue because drug treatments often affect the levels of typical housekeeping proteins such as actin or tubulin.” We have used actin as well as Coomassie blue as loading controls in cases involving shRNA knockdown or overepxpression of specific proteins (no drug treatments), which was not done in this manuscript.
- The paragraph lengths in some sections, such as the discussion, are inconsistent—either too long or too short. Authors are advised to maintain uniformity for better readability.
Thank you for your advice. In section 3.4, we added paragraph 4 (yellow highlight) to paragraph 3 and paragraph 6 (yellow highlight) to paragraph 5 to eliminate paragraphs that are too short. Similarly, in section 3.5, we added paragraph 3 (yellow highlight) to paragraph 2. In the revised discussion, we expanded the “future studies” section and add a “limitations” section to improve readability.
- Authors need to specify if the combination of MIA-602/690 and PI3K inhibitors overcome resistance mechanisms in prostate cancer with PTEN loss or PIK3CA mutations more effectively than current therapies.
Thank you for the suggestion. The PCa cell lines with known PTEN loss of function are LNCaP, PC3, and H660 (Table 1). There is a suggestion in PTEN deficient PCa that PI3Ka activity is suppressed and PI3Kb drives PI3K signaling (ref 48). Except for LNCaP, we did not find a correlation in PTEN deficient PCa (PC3, H660) for PI3Kbi in the MIA-602/690 + PI3K inhibitor combination. The only known PIK3CA mutation occurs in 22Rv1 (Table 1). In this case, MIA-602/690 worked better with PI3Kai. MIA-602/690 + PI3K isoform inhibitors is more effective than enzalutamide alone in LNCaP (60% vs 12% cell death). We add to the future studies section of the discussion “investigate mechanisms of potential resistance mediated by MIA602/690 + PI3K isoform inhibitors”.
- Authors need to discuss the potential off-target effects or toxicity concerns of combining GHRH antagonists with PI3K inhibitors in preclinical and clinical settings.
Thank you for the suggestion. Extensive investigations of MIA-602/690 in mouse models have not found any toxic side effects; clinical testing has not yet started. PI3K isoform inhibitors are less toxic compared to pan-PI3K inhibitors; however, some toxicity issues remain. We added the sentence in the discussion “future studies will determine whether MIA-602/690 (less toxic) can lower the dose of PI3K isoform inhibitors in order to reduce toxicity without affecting anti-tumor efficacy.”
- Discussion section needs to avoid repetition of result section.
Thank you for the suggestion. We have made some changes to the discussion to avoid repetition of results.
- A deeper discussion of study limitations, challenges, and future research directions, along with a more detailed introduction and discussion, would enhance the manuscript's completeness. To address this deficiency and align with the relevant findings presented in the manuscript, the authors are encouraged to integrate pertinent literature (please refer to PMID: 39505776, 35895804, 38965223). This literature notably delves into the modifications of critical signaling pathways, elucidating alterations in growth in prostate cancer including GHRH. By integrating these studies into the manuscript, the authors can substantially strengthen the basis for their compelling and comprehensive concept of GHRH/PI3K axis in regulation of prostate cancer.
Thank you for the suggestions. We have added a “limitations of this study” and expanded “future studies” in the discussion. We have incorporated PMID: 39505776 (GHRH and the prostate; ref. 36) and discussed the potential link between GHRHR and RTK/ERK signaling relevant to our data. In the conclusion section of the discussion, we stated “PI3K inhibitors should further block the RTK/ERK signaling pathway (shared PI3K family members) in AR negative CRPC/NEPC above the RTK/ERK inhibition resulting from GHRH antagonists”. A schematic of this proposed link is presented in Figure 7.
- The manuscript requires a significant attention to improve punctuations, grammar and the readability specifically in discussion section.
Thank you for the suggestion. We have re-reviewed the manuscript for improvements in grammar, conciseness, and readability, and made the appropriate changes.
Reviewer 3 Report
Comments and Suggestions for Authors
This manuscript shows the effect of growth hormone-releasing hormone (GHRH) in prostate cancer. They show the clinical implication with GHRH and GHRH receptor (GHRHR) expressions in prostate cancer patients. They performed the biological analyses with GHRH inhibitor suggesting, suppression of GHRH induces apoptosis and decreasing of PI3K signaling. Their findings are novel and interesting; however, the overall quality of the experimental data is poor. It would be difficult to draw the conclusion with poor data in the presented manuscript. There are three major concerns for the publication. These concerns can be addressed with additional experiments.
Comments
1. The interpretation of signal analysis by Western blotting is probably wrong. PI3K signaling is mainly regulated by phosphorylation by kinase activity; however, some of their data show loss of total protein amount of signal proteins. It is also unclear which phosphorylation site is being detected. Expression of cell cycle-related genes are not detected in some of samples, which would suggest that the cell proliferation should have completely stopped. But the authors evaluate only the apoptosis. The pathways remain unclear.
2. Most panels show multiple samples in the Figure 2-4, but the authors only perform the t-test. This is an incorrect statistical test, and these analyses need to be reperform all analysis with the appropriate test.
3. Some prostate cancer cell lines selected for the analysis in the manuscript, but the expression of GHRHR is unclear.
Comments on the Quality of English Languageplz see the comments.
Author Response
- The interpretation of signal analysis by Western blotting is probably wrong. PI3K signaling is mainly regulated by phosphorylation by kinase activity; however, some of their data show loss of total protein amount of signal proteins. It is also unclear which phosphorylation site is being detected. Expression of cell cycle-related genes are not detected in some of samples, which would suggest that the cell proliferation should have completely stopped. But the authors evaluate only the apoptosis. The pathways remain unclear.
Thank you for the comments. We used antibodies that detect total PI3Ka and PI3Kb from Cell Signaling, which is not specific to any phosphorylation site. However, we agree that a more detailed and in-depth analysis must investigate the consequences of how the MIA-602/690 + PI3K isoform inhibitor combination (and each alone) alters the phosphorylation sites of PI3Ka and PI3Kb.
To measure effects on proliferation, we used the CellTiter Aqueous colorimetric method. Previously, we referred this as a “viability assay” but believe it is more accurate as a “proliferation assay”. Results presented in Supplementary Figure S8 suggest proliferation did not completely stop (greatest inhibition of proliferation at 72h was 31% of control (=100%). However, we agree that the specific mechanistic pathways remain unclear and add this to the “limitations” section of the discussion.
- Most panels show multiple samples in the Figure 2-4, but the authors only perform the t-test. This is an incorrect statistical test, and these analyses need to be reperform all analysis with the appropriate test.
Thank you for your advice. We used the t-test to compare 2 samples directly. For example, MIA-602 + PI3K isoform inhibitor compared to MIA-602 alone and to PI3K isoform inhibitor alone. In the revised manuscript, we used one-way ANOVA followed by Dunnett’s or Šidáks multiple comparisons to recalculate Figures 2-4 and the P values remain similar to what was obtained by the t-test. This analysis is presented as Supplementary Figure S2.
- Some prostate cancer cell lines selected for the analysis in the manuscript, but the expression of GHRHR is unclear.
In Supplementary Figure S6, we compared the GHRHR protein levels in LNCaP, 22Rv1, PC3, H660, LASCPC compared to the non-cancer RWPE-1 by Western blot. Results suggested a slightly higher GHRHR in PCa cell lines compared to RWPE-1, with PC3 containing high levels of a splice variant (SV1).
Reviewer 4 Report
Comments and Suggestions for Authors
Significance: GHRH expression in cancers has led to the development of peptide 27 antagonists (e.g., MIA-602 and -690) for therapeutic treatment of PCa. While in vitro these competitive blockers of the GHRHR showed efficacy, their utility in the clinics has proven disappointing as single agents, particularly for mCRPC and the NEPC subtype. Therefore, the authors attempted to identify combinations with existing compounds that would improve the killing (or viability) of a panel of PCa cell lines – a strategy that is not novel by any stretch.
From its conceptualization to its execution the overall study is significantly flawed for the reasons detailed below (not exhaustive).
- Figure 1 portends higher expression of GHRHR and GHRH mRNA in PCa compared to normal tissue (weak correlation) and a modest DFS outcome. It is not entirely clear what makes this figure novel nor that significant as a justification for their study. It is in fact fairly gratuitous, as all of their studies are conducted in vitro from that point on with cell lines that only superficially reflect human PCa, and none of their studies have been conducted in animals.
- There is no rationale that I can see in selecting particular drug combination for their studies with MIA synthetic lethality.
- 2-4 are all trypan blue exclusion experiments to monitor the ratio of live/dead cells, but none of these are complemented with ‘viability assays’ to determine if there are still cells capable of proliferation – some of these are eventually included in SI (but only for specific experiments that are also questionable).
- ‘The AKT1-3 inhibitor AZD5363 (capivasertib) or the mTOR inhibitor rapamycin 276(both are downstream mediators of PI3K signaling [22,25-27]), had little or no effect at 277increasing cell death + MIA-602 or -690 in DU145, H660, and LASCPC (Supplementary 278Figure S3 and S4). Similarly, the NFB inhibitor parthenolide did not increase MIA-602 or 279-690 cell death in LASCPC (Supplementary Figure S4).’ SO, what is the mechanism then? Why would PI3K inhibitors help, particularly in PTEN-null cell lines?
- The multiple panels WBs (Fig. 5) are very puzzling if not questionable. In particular, the result of induced AR (rapid and complete) loss by 602 or 690 + βi in LNCaP cells (PTEN null) seems inexplicable. In fact, it was not previously identified in a landmark Cancer Cell paper by Carver and Sawyer (10.1016/j.ccr.2011.04.008). It is to be assumed that there are normally sufficient levels of GHRH in conditioned medium containing 5% FCS. The most logical explanation for such result is that most of the cells were dead (as suggested by the presence of Cl-PARP). But if they are dead, how acceptable are the WBs in general?
- SI7. The authors somehow concluded that MIA+PI3K αi or βi have synergistic effects on viability (Aqueous 96). I can’t comment on the analysis since the calculations are not shown, but a visual inspection does not support this. Also, if they used Tallarida methods, it can’t be explained why going from 2.5 to 5 µM (two-fold dilution) there would be such a huge increase in P value (that is not visible of the graph). The figure is also poorly detailed and does not explain after how many days was viability assayed.
- ‘Because of the unexpected result that MIA-602 and -690 greatly reduced AR in 371LNCaP and 22Rv1 (Figure 5B, C and Figure 6B, C), we investigated whether adding the 372AR antagonist enzalutamide can improve efficacy. Results indicated that enzalutamide 373did not enhance cell death mediated by MIA-602/690 in LNCaP ‘. This reasoning is very obscure! If the AR is ‘gone’ why would they expect that ENZ would make any difference?
- As could be expected from a ‘weak’ paper, the Discussion fails to make any solid conclusions from the study, or explain what might the mechanism(s) for their (crude) observations be.
Author Response
- Figure 1 portends higher expression of GHRHR and GHRH mRNA in PCa compared to normal tissue (weak correlation) and a modest DFS outcome. It is not entirely clear what makes this figure novel nor that significant as a justification for their study. It is in fact fairly gratuitous, as all of their studies are conducted in vitro from that point on with cell lines that only superficially reflect human PCa, and none of their studies have been conducted in animals.
Thank you for the constructive criticism. Our rationale in Fig. 1 was to suggest that GHRHR and GHRH were only slightly higher in PCa compared to normal tissue. To the best of our knowledge, this analysis has not been previously published. In section 3.1, we add the sentence “This weak correlation provided some justification to investigate combinations that enhance GHRH antagonists as a therapeutic strategy.”
We add a “limitations of this study” section in the discussion stating, “a lack of testing the MIA-602/690 + PI3K isoform inhibitor combination in a mouse model of CRPC/NEPC”.
- There is no rationale that I can see in selecting particular drug combination for their studies with MIA synthetic lethality.
Thank you for the intelligent observation. To initiate this study, we used inhibitors available in the lab that targeted different signaling pathways (apoptosis, antimitotic, cell cycle, proteasome). Our goal was to identify a drug that significantly enhanced MIA602 and -690 cell death in LNCaP, 22Rv1, and PC3 greater than each drug alone. Only when we used the pan-PI3K inhibitor LY294002 did we achieve this threshold. This led to further experiments using the PI3K isoform inhibitors.
- 2-4 are all trypan blue exclusion experiments to monitor the ratio of live/dead cells, but none of these are complemented with ‘viability assays’ to determine if there are still cells capable of proliferation – some of these are eventually included in SI (but only for specific experiments that are also questionable).
To measure effects on proliferation, we used the CellTiter Aqueous colorimetric method. Previously, we referred this as a “viability assay” but believe it is more accurate as a “proliferation assay”. Results presented in Supplementary Figure S8 for LNCaP and PC3 indicated MIA602/690 + PI3K isoform inhibitor decreased cell proliferation but not completely (greatest inhibition of proliferation at 72h was 31% of the control (=100%).
- ‘The AKT1-3 inhibitor AZD5363 (capivasertib) or the mTOR inhibitor rapamycin 276(both are downstream mediators of PI3K signaling [22,25-27]), had little or no effect at 277increasing cell death + MIA-602 or -690 in DU145, H660, and LASCPC (Supplementary 278Figure S3 and S4). Similarly, the NFkB inhibitor parthenolide did not increase MIA-602 or 279-690 cell death in LASCPC (Supplementary Figure S4).’ SO, what is the mechanism then? Why would PI3K inhibitors help, particularly in PTEN-null cell lines?
Thank you for the questions. At this time, we do not have a specific mechanism and state this in the “limitations of this study” section of the discussion. We suggest that MIA-602/690 alters multiple signaling pathways (apoptosis, cell proliferation, PI3Ka/b, AKT, ERK, and AR) and this may weaken activation of adaptive mechanisms resulting from inhibition of PI3K and improve efficacy. Further investigations are required to determine if this is correct.
The PCa cell lines with known PTEN loss of function are LNCaP, PC3, and H660 (Table 1). There is a suggestion in PTEN deficient PCa that PI3Ka activity is suppressed and PI3Kb drives PI3K signaling (ref 47). Except for LNCaP, we did not find a correlation in PTEN deficient PCa (PC3, H660) for PI3Kbi in the MIA-602/690 + PI3K inhibitor combination.
- The multiple panels WBs (Fig. 5) are very puzzling if not questionable. In particular, the result of induced AR (rapid and complete) loss by 602 or 690 + βi in LNCaP cells (PTEN null) seems inexplicable. In fact, it was not previously identified in a landmark Cancer Cell paper by Carver and Sawyer (10.1016/j.ccr.2011.04.008). It is to be assumed that there are normally sufficient levels of GHRH in conditioned medium containing 5% FCS. The most logical explanation for such result is that most of the cells were dead (as suggested by the presence of Cl-PARP). But if they are dead, how acceptable are the WBs in general?
We agree that the result with MIA-602/690 reducing AR was unexpected and this has not been previously reported. PI3K inhibitors had no effect on AR. At this time, we do not know the mechanism of the AR reduction by MIA-602/690 and further investigations are required. We do not agree that most of the cells are dead, and this may be a reason for decreased AR. From Fig. 2, treatment of LNCaP with MIA-602/690 resulted in only 16-26% cell death at 72h and AR was drastically reduced (Fig. 5C).
- The authors somehow concluded that MIA+PI3K αi or βi have synergistic effects on viability(Aqueous 96). I can’t comment on the analysis since the calculations are not shown, but a visual inspection does not support this. Also, if they used Tallarida methods, it can’t be explained why going from 2.5 to 5 µM (two-fold dilution) there would be such a huge increase in P value (that is not visible of the graph). The figure is also poorly detailed and does not explain after how many days was viability assayed.
We used the CalcuSyn software (Chou method) to calculate the combination index (CI) for the MIA-602Ac and -690Ac + PI3Kbi (LNCaP) or PI3Kai (PC3). Unfortunately, there was a registration issue with the software that was not resolved until after submission. We have completed the analysis and presented the data in Supplementary Table S1 (located after Supplementary Figure S8). The results suggested a strong synergism with CI values ranging from 0.11 to 0.39.
- ‘Because of the unexpected result that MIA-602 and -690 greatly reduced AR in 371LNCaP and 22Rv1 (Figure 5B, C and Figure 6B, C), we investigated whether adding the 372AR antagonist enzalutamide can improve efficacy. Results indicated that enzalutamide 373did not enhance cell death mediated by MIA-602/690 in LNCaP ‘. This reasoning is very obscure! If the AR is ‘gone’ why would they expect that ENZ would make any difference?
We add in section 3.6 “Our reasoning was that if AR levels were reduced, addition of enzalutamide would be more effective due to the presence of less AR target. This rationale was previously supported by using a deubiquitinase inhibitor (betulinic acid) to reduce AR and adding enzalutamide to enhance apoptotic cell death in LNCaP [38].”
MIA-602/690 in TFA form did result in rapid loss of AR (Fig. 5C) but subsequent use of MIA-602/690 in Ac form resulted in a less dramatic decrease in AR (Fig. 6B).
- As could be expected from a ‘weak’ paper, the Discussion fails to make any solid conclusions from the study, or explain what might the mechanism(s) for their (crude) observations be.
In the revised discussion, we add “Limitations of this study are a lack of identifying a specific mechanism why PI3K isoform inhibitors enhance MIA-602/690 cell death”. We suggest that MIA-602/690 alters multiple signaling pathways (apoptosis, cell proliferation, PI3Ka/b, AKT, ERK, and AR) and this may weaken activation of adaptive mechanisms resulting from inhibition of PI3K and improve efficacy. Further investigations are required to determine if this is correct.
Round 2
Reviewer 3 Report
Comments and Suggestions for Authors
The authors addressed all my concerns.
Author Response
Thank you.